# A Concise Review on Oxidative Stress-Mediated Ferroptosis and Cuproptosis in Alzheimer’s Disease

**DOI:** 10.3390/cells12101369

**Published:** 2023-05-12

**Authors:** Xudong Huang

**Affiliations:** Neurochemistry Laboratory, Department of Psychiatry, Massachusetts General Hospital and Harvard Medical School, Charlestown, MA 02129, USA; huang.xudong@mgh.harvard.edu; Tel.: +1-617-724-9778

**Keywords:** Alzheimer’s disease, aging, oxidative stress, iron, ferroptosis, copper, cuproptosis

## Abstract

Alzheimer’s disease (AD), which was first identified more than a century ago, has become a pandemic that exacts enormous social burden and economic tolls as no measure of combating devastated AD is currently available. Growing etiopathological, genetic, and biochemical data indicate that AD is a heterogeneous, polygenic, multifactorial, and complex disease. However, its exact etiopathology remains to be determined. Numerous experimental data show that cerebral iron and copper dyshomeostasis contribute to Aβ amyloidosis and tauopathy, two neuropathological hallmarks of AD. Moreover, increasing experimental evidence suggests ferroptosis, an iron-dependent and nonapoptotic form of cell death, may be involved in the neurodegenerative process in the AD brain. Thus, the anti-ferroptosis approach may be an efficacious therapeutic strategy for AD patients. Furthermore, it remains to be further determined whether cuproptosis, a copper-dependent and distinct form of regulated cell death, also plays a contributing role in AD neurodegeneration. We hope this concise review of recent experimental studies of oxidative stress-mediated ferroptosis and cuproptosis in AD may spur further investigations on this timely and essential line of research.

## 1. Introduction

In 1906, at the meeting of the Southwest German Psychiatrists in Tubingen, Germany, a German psychiatrist and neuropathologist, Dr. Alois Alzheimer, from the Munich University Hospital, presented a clinical case of a female patient, 50-year-old Auguste Deter [1]. He followed her for five years from her admission for progressive sleep disturbance, cognitive impairment, psychosocial dysfunction, and behavioral changes such as paranoia, aggression, hallucinations, delusion, confusion, etc., until her death [1]. The later published neurohistological findings, unique plaques and neurofibrillary tangles (NFTs) in her postmortem brain, are now recognized as brain β-amyloid (Aβ) plaques and microtubule-associated and hyperphosphorylated protein tau (τ) in the paired helical filaments (PHFs) [2], as shown in Figure 1. Ever since, the most common cause of age-related dementia, Alzheimer’s disease (AD), has been named after Dr. Alzheimer, and Aβ amyloidosis, together with tauopathy, has become a neuropathological hallmark for AD. Nevertheless, more than 200 AD clinical research programs and clinical trials of experimental therapeutics against both Aβ amyloid directly and its related targets, protein τ, and other targets, have failed so far. Thus, this necessitates a deeper and more thorough understanding of the pathogenic mechanisms of AD. Moreover, growing etiopathological, genetic, and biochemical data indicate that AD is a heterogeneous, polygenic, multifactorial, and complex disease. However, its exact etiopathology remains to be determined.

Numerous experimental data have shown that brain iron and copper have been dysregulated in AD [3]. Recent data indicate that ferroptosis and cuproptosis, iron- and copper-dependent nonapoptotic forms of cell death, may also be involved in AD neurodegeneration. Hence, they may represent potential targets for AD therapeutic interventions. Thus, it is necessary to evaluate recent experimental data from this line of research. In this short review manuscript, we have summarized some current studies. Notably, we have tabulated some recent and selected studies on anti-ferroptosis agents as potential AD therapeutics using relevant cellular and animal models.

## 2. Alzheimer’s Disease

According to dementia statistics by the World Health Organization (WHO), there are more than 55 million people currently living with dementia and nearly 10 million new cases every year worldwide. AD is the most common senile dementia, accounting for 60–70% of all dementia cases [4]. In the US alone, approximately 6.7 million Americans aged 65 and older are living with AD. Its prevalence is projected to be 13.8 million by 2060 due to the burgeoning older population barring the quantum leap in preventing, slowing, or curing AD. It has become a pandemic that exacts enormous social burden and economic tolls as no measure of combating devastated AD is currently available [5].

AD commences as insidious short-term memory debilitation and presents with spatiotemporal disorientation and irreversible cognitive deterioration. It gradually worsens other cognitive functions rendering patients uncommunicative and non-ambulatory after 10–15 years of disease progression. AD is a multifactorial and genetically complex disease, and most AD cases are sporadic or late-onset, and their risk is predominantly controlled by the APOE genotype [5]. The APOE ε4 allele increases AD risk in allele-dose dependent fashion, while the ε2 allele has a risk-mitigating effect [6,7]. Early-onset familial AD (FAD) accounts for 2–5% of all AD cases, and it is caused by mutations within the amyloid precursor protein (APP) [8,9,10] or presenilin 1 (PS1) [11] or 2 (PS2) [12] genes, which are inherited in an autosomal dominant fashion. In addition, neuropathological hallmarks of AD include the accumulation of insoluble Aβ peptides in the form of parenchymal plaques and vascular deposits, intraneuronal NFTs composed of misfolded and hyperphosphorylated microtubule-associated τ protein in the PHFs. Other neuropathological features include activated astrocytes and microglia, blood-brain barrier (BBB) degradation, neuroinflammation, oxidative stress, brain metal dysregulation, widespread loss of synapses, impaired cerebral energy metabolism, and neuronal demises. Figure 2 summarizes major neuropathological features of AD.

Aβ peptide as the main constituent of Aβ plaques and vascular deposits was first identified by the late Dr. George Glenner in 1984 [13] and later by Drs. Konrad Beyreuther and Colin Masters in 1985 [14]. Aβ peptides that have 38–43 residues [8] are generated by proteolysis of a much larger metalloprotein- APP, a type I membrane protein [9,10]. APP is expressed ubiquitously and, in the amyloidogenic pathway of APP processing, β-site cleaving enzyme 1 (BACE1) or β-secretase initially cleaves APP to form the N-terminus of Aβ at the Asp+1 residue of the Aβ sequence. Two cleavage products are formed: a secreted ectodomain of APP termed sAPP-β and the membrane-bound C-terminal fragment β (CTF-β) or C99. The CTF-β fragment is cleaved by γ-secretase to produce the C-terminus of the Aβ peptide and an APP intracellular fragment (AICD). A majority of the Aβ peptides generated by γ-secretase end at amino acid 40 (Aβ40), and a small proportion end at amino acid 42 (Aβ42). Alternatively, in the non-amyloidogenic pathway of APP processing, APP cleavage by α-secretase occurs within the Aβ domain at Leu+17 and produces a secreted sAPP-α ectodomain, and a CTF-α or C83, which is then subsequently cleaved by γ-secretase to form the AICD and non-amyloidogenic three kDa fragment, P3. Other less frequent APP cleavage pathways such as η- and δ-secretase cleavage pathways are not presented here [15]. Figure 3 depicts major amyloidogenic and non-amyloidogenic cleavage pathways of APP.

Understanding APP biology has led to the development of the Alzheimer’s Aβ cascade hypothesis, which proposes the accumulation of Aβ as a primary culprit of AD pathogenesis and a critical therapeutic target [16]. Building on this premise, the development of Aβ-directed immunotherapeutics and inhibitors of APP proteases: β-secretase and γ-secretase, have been pursued over the past two decades. However, a series of setbacks these approaches encountered during clinical trial testing hampered progress toward successful clinical development. In June 2021, the FDA controversially approved a monoclonal antibody drug called Aducanumab (marketed as Aduhelm) with marginal cognitive benefit [17], using its accelerated approval pathway. Biogen and Eisai have jointly developed Aducanumab, which is based upon its amyloid clearance effects. Nevertheless, another monoclonal antibody called Lecanemab (marketed as Leqembi), which has also been jointly developed by the same companies based on the same scientific premise, also gained accelerated FDA approval early this year. It has shown a moderate cognitive benefit for AD patients in a recent phase III clinical trial [18]. However, due to its looming safety concerns, a complete risk-benefit analysis and additional clinical evidence of Lecanemab may be needed before its Medicare coverage in the U.S. Some AD patients taking monoclonal antibody drugs, such as Aducanumab, Lecanemab, etc., have developed amyloid-related imaging abnormalities (ARIA). ARIA is demonstrated as brain edema or sulcal effusion (ARIA-E) and/or as hemosiderin deposits caused by brain parenchymal or pial hemorrhage (ARIA-H) [19].

Tauopathy also plays a critical role in AD neuropathogenesis. In its native state, tau protein is an unfolded and highly soluble protein that is responsible for the assembly of tubulin and stabilization of microtubules, promoting normal neuronal function. In AD, its randomly coiled structure becomes a β-sheet conformation, favoring tau protein misfolding and aggregation. Further, its most common and notable post-translationally modified form is the hyperphosphorylated tau (p-tau). It is believed to be the key driver for the formation of intracellular NFTs, another neuropathological hallmark of AD as shown in the following Figure 4 [20,21]. More interestingly, recent AD genetics studies have demonstrated that an autosomal dominant AD PS1 E280A mutation carrier did not develop mild cognitive impairment (MCI) until her seventies, three decades after the expected age of clinically early AD onset. This individual, who had unusually high cerebral Aβ amyloid burden and limited tauopathy and neurodegeneration features, also happened to have APOE ε3 Christchurch (R136S, APOEε3Ch) variant homozygosity. In vivo follow-up PET imaging and postmortem studies indicated that her frontal cortex and the hippocampus (the most affected brain regions in AD) waere less affected than the occipital cortex by tau pathology [22,23]. Thus, cerebral Aβ amyloidosis alone may not be sufficient to induce full bloom AD pathology. So far, experimental drugs in various development pipelines against tauopathy have not shown clinical success.

## 3. Ferroptosis and Alzheimer’s Disease

Numerous experimental data indicate that brain iron dysregulation is involved in Alzheimer’s pathogenesis [24]. Our early in vitro study has first demonstrated that Aβ can directly reduce Fe^3+^ to Fe^2+^, initiates the Fenton reaction, and engenders oxidative stress via a heightened generation of reactive oxygen species (ROS) such as H_2_O_2_ and HO• radical production [3,25]. We first employed synchrotron-based X-ray fluorescence microscopy (XRFM) and detected abnormal iron enrichment in procured Aβ amyloid plaques from AD postmortem brain tissues using the laser capture microdissection (LCM) technique [26]. The Aβ- Fe^3+^ redox interaction mechanism has been validated in Aβ amyloid plaque cores from AD postmortem and transgenic mouse brain tissues [27,28,29,30,31]. More recently, even nanoscale metallic iron (Fe^0^), together with magnetite (Fe_3_O_4_), Fe^2+^, and Fe^3+^, have been identified in Aβ amyloid plaque cores from AD postmortem brain tissue [32]. Interestingly, magnetite pollutant nanoparticles (diameter < 200 nm) have been shown to enter the human brain directly via the olfactory bulb [33].

Ferroptosis is an iron-dependent and nonapoptotic form of cell death first identified in cancer cells, and its inhibition may deter neurodegeneration [34]. It is characterized by glutathione depletion due to cystine uptake inhibition by the cystine/glutamate transporter (X_c_^−^ or xCT) [34], impaired activity of the selenoprotein– glutathione (GSH) peroxidase 4 (GPX4), and increased lipid peroxidation [35]. Figure 5 shows the ferroptosis signaling pathway and major ferroptosis-interdicting strategies: iron chelation, antioxidation, and oxidized lipid removal by GPX4.

The ferroptosis associated with cerebral iron dysregulation has been hypothesized and suggested as a unique neurodegenerative mechanism for neurologic diseases, including AD [24,37,38,39,40,41]. The hypothesis of ferroptosis in AD pathology seems to gain support from investigations in clinical samples, AD transgenic animal models, and cell line studies. If validated, the anti-ferroptosis approach may be an efficacious therapeutic strategy for AD patients.

Indeed, brain iron dyshomeostasis, upregulated xCT (impaired glutathione metabolism), and lipid peroxidation, the pathological features of ferroptosis, have recently been demonstrated in AD brain samples compared to cognitively normal ones [42]. Protein levels of NADPH oxidase 4 (NOX4) were significantly elevated in impaired astrocytes of AD patients, and APP/PS1 AD transgenic mouse models, and its elevation increased ferroptosis-dependent cytotoxicity by activating oxidative-stress-induced lipid peroxidation in human astrocytes. These data suggest that NOX4 promotes the ferroptosis of astrocytes by oxidative-stress-induced lipid peroxidation via the impairment of mitochondrial metabolism in AD [43].

Ferroportin1 (Fpn) is the only identified mammalian nonheme iron exporter, and it was downregulated in both the brains of APP/PS1 AD transgenic mice and patients. Bao et al. have demonstrated that restoring Fpn ameliorated ferroptosis and memory impairment in AD mice, indicating the critical role of Fpn and ferroptosis in AD pathogenesis [44]. Huang et al. used a triple omics approach that integrated transcriptomic, proteomic, and metabolomic data collected from the MC65 human nerve cell line (SK-N-MC human neuroblastoma parent cell line) to express the C99 fragment of APP. They have demonstrated that the toxic intracellular aggregation of Aβ induced oxytosis/ferroptosis regulated cell death [45]. Studies have also shown that the presenilins (PS1 and 2) promote the expression of GPX4, and their mutations sensitize cells to ferroptosis by suppressing GPX4 expression [46].

As summarized in Table 1, there have been many efforts to identify anti-ferroptosis agents as potential AD therapeutics. A recent study has shown that seafood-derived plasmalogens (PLs) can partly improve the swimming performance in the AlCl_3_-exposed AD zebrafish model by suppressing neuronal ferroptosis and accelerating synaptic transmission at the transcriptional level. This result indicates that PLs can be developed as a functional food supplement to relieve AD symptoms [47]. Another study has shown that tetrahydroxy stilbene glycoside (TSG) ameliorates AD pathogenesis in APP/PS1 AD transgenic mice via the activation of GSH/GPX4/ROS and Keap1/Nrf2/ARE signaling pathways and suppression of related ferroptosis [48]. In addition, a comprehensive library of >900 natural compounds were screened for protection against oxytosis/ferroptosis using HT22 mouse hippocampal nerve cells and MC65 cells to understand better the chemical nature of inhibitors of oxytosis/ferroptosis. A small set of potent anti-oxytosis/ferroptosis compounds highly enriched in plant quinones was identified. It appears that the pro-oxidant character of a quinone compound can coexist with an inhibitory effect on lipid peroxidation and, consequently, still prevent oxytosis/ferroptosis [49]. Moreover, eriodictyol was found to ameliorate cognitive impairment in APP/PS1 AD transgenic mice by interdicting ferroptosis via vitamin D receptor mediated Nrf2 activation [50]. Further, using male APP/PS1 AD transgenic mice, Aβ42-exposed N2a cells, erastin-stimulated HT22 cells (a cellular model for ferroptosis), and LPS-induced BV2 cells, forsythoside A (FA) has been demonstrated to mitigate AD pathology by inhibiting ferroptosis-mediated neuroinflammation via Nrf2/GPX4 axis activation [51]. Furthermore, senegenin (Sen) has recently exhibited intense neuroprotective activity against Aβ25-35-induced oxidative damage and the lipid metabolic associated with ferroptosis in PC12 cells [52]. Finally, iron chelator- deferoxamine ameliorated aluminum-maltolate-(Al(mal)_3_)-induced neuronal ferroptosis and attenuated oxidative stress in adult rats [53].

## 4. Cuproptosis and Alzheimer’s Disease

Experimental evidence has shown that cerebral copper dyshomeostasis and neurotoxicity are also implicated in Alzheimer’s pathology [54]. However, a recent community-based study suggests cerebral copper may protect AD patients from cognitive decline. It may also imply that dietary copper supplementation may slow AD patients’ cognitive decline [55]. Nevertheless, a recent study has indicated that chronic copper exposure leads to hippocampus oxidative stress and impaired learning and memory in male and female rats [56]. Furthermore, a meta-analysis of case-control studies suggests a positive relationship between serum copper levels and AD risk [57]. Given the study discrepancy in copper roles in AD pathology, we believe the chemical speciation of copper ion (in Cu^2+^, Cu^+^, or mixed form) in the brain may be more relevant to AD pathology than its absolute concentration, as redox-active copper may promote cerebral oxidative stress.

Our early study indicated that Aβ-Cu^2+^ redox interaction engenders Cu^2+^ reduction to Cu^+^ and cell-free H_2_O_2_ production via the Haber–Weiss reaction [3,58]. Indeed, nanoscale metallic copper (Cu^0^) and Cu^+^ and Cu^2+^ have also recently been found in Aβ amyloid plaque cores from AD postmortem brain tissue [32]. Moreover, low levels of copper exposure in drinking water disrupted brain Aβ amyloid homeostasis by altering its production and clearance in AD transgenic mice overexpressing APP [59]. Interestingly, cuproptosis, a copper-dependent and distinct form of regulated cell death, has recently been discovered to occur utilizing the direct binding of copper to lipoylated components of the tricarboxylic acid (TCA) cycle. It results in lipoylated protein aggregation and subsequent iron–sulfur cluster protein loss, leading to proteotoxic stress and cell death [60]. Figure 6 depicts the cuproptosis mechanism and key cuproptosis-inhibiting strategies: copper chelation, antioxidation, and copper removal from cells by upregulating copper exporters such ATP7A/B.

More intriguingly, Lai et al. identified two cuproptosis-related molecular gene clusters using 310 AD clinical samples, a weighted correlation network analysis (WGCNA) algorithm, and machine learning (ML) tools. These include elevated immune responses and immune infiltration that are highly associated with the Aβ42 levels and β-secretase activity [62]. Moreover, exposure of Cu in drinking water induces cognitive impairment by mediating cuproptosis, damaging synaptic plasticity, and inhibiting the CREB/BDNF pathway in mice [63]. Furthermore, using a bioinformatics approach, seven AD-related cuproptosis hub genes and six drugs targeting these genes have been identified from the Gene Expression Omnibus database and DrugBank, respectively [64]. In our own studies, we have further validated the exacerbation of Aβ oligomerization due to Cu^2+^/Aβ/H_2_O_2_ redox interactions in vitro. We have also reported that dietary Cu exposure enhanced APP translations via its 5′ untranslated region (5′UTR) of mRNA in human SH-SY5Y cells, increased Aβ amyloidosis, and heightened levels of associated pro-inflammatory cytokines such as MCP-5 in APP/PS1 AD transgenic mice [65]. In summary, it remains to be further determined whether cuproptosis also plays a contributing role in AD neurodegeneration.

## 5. Conclusions

Most experimental data suggests that cerebral dysregulation of redox-active iron and copper may contribute to AD pathogenesis, though some data discrepancies among different studies need to be further addressed. Moreover, ferroptosis and cuproptosis (if verified) may also be distinctive neurodegenerative mechanisms of AD. Furthermore, this line of research may yield potentially druggable targets for developing promising AD therapeutics.

## Figures and Tables

**Figure 1 cells-12-01369-f001:**
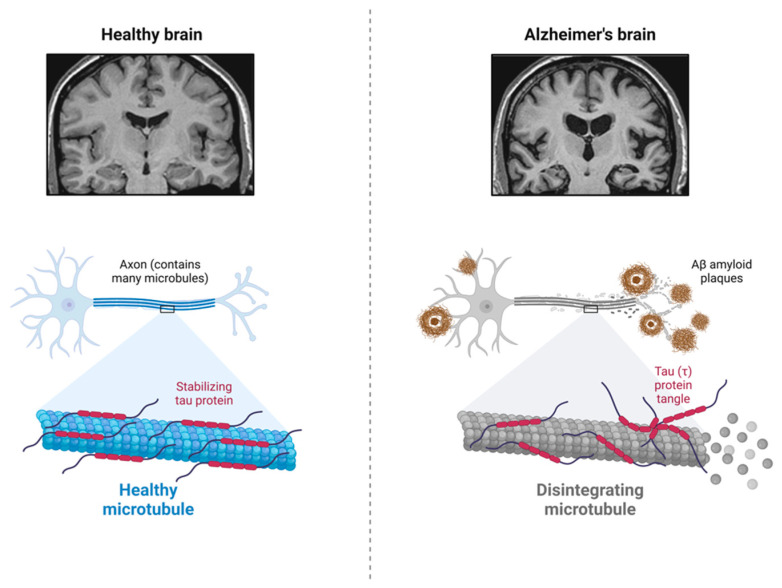
Major neuropathological hallmarks of Alzheimer’s disease. AD brain usually contains Aβ amyloid plaques and neurofibrillary tangles (NFTs), and the microtubule-associated and hyperphosphorylated protein tau (τ) in the paired helical filaments (PHFs). Created with BioRender.com.

**Figure 2 cells-12-01369-f002:**
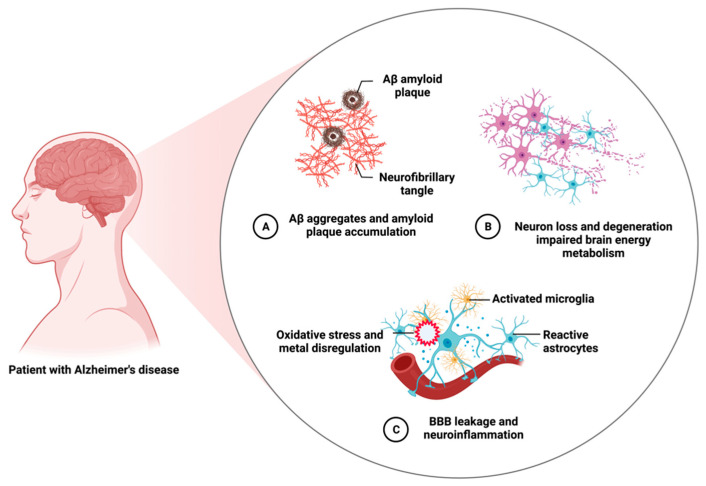
Salient neuropathological features of Alzheimer’s disease. AD is a heterogeneous, polygenic, multifactorial, and complex disease. It exhibits many neuropathological features other than its major hallmarks. Created with BioRender.com.

**Figure 3 cells-12-01369-f003:**
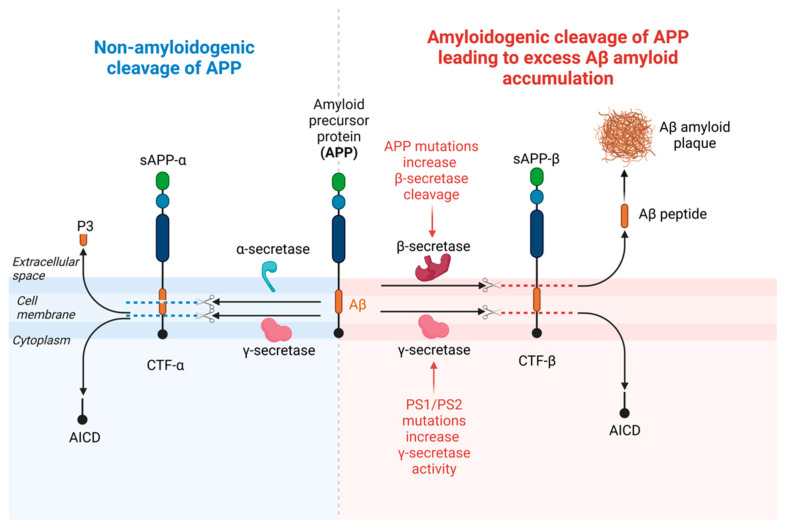
Schematic illustration of APP major cleavage pathways by secretase enzymes (adapted from [15]). In the amyloidogenic pathway of APP processing, β-site cleaving enzyme 1 (BACE1) or β-secretase initially cleaves APP to form the N-terminus of Aβ at the Asp+1 residue of the Aβ sequence. Two cleavage products are formed: a secreted ectodomain of APP termed sAPP-β and the membrane-bound C-terminal fragment β (CTF-β) or C99. The CTF-β fragment is cleaved by γ-secretase to produce the C-terminus of the Aβ peptide and an APP intracellular fragment (AICD). A majority of the Aβ peptides generated by γ-secretase end at amino acid 40 (Aβ40), and a small proportion end at amino acid 42 (Aβ42). Alternatively, in the non-amyloidogenic pathway of APP processing, APP cleavage by α-secretase occurs within the Aβ domain at Leu+17 and produces a secreted sAPP-α ectodomain, and a CTF-α or C83, which is then subsequently cleaved by γ-secretase to form the AICD and non-amyloidogenic three kDa fragment, P3. Created with BioRender.com.

**Figure 4 cells-12-01369-f004:**
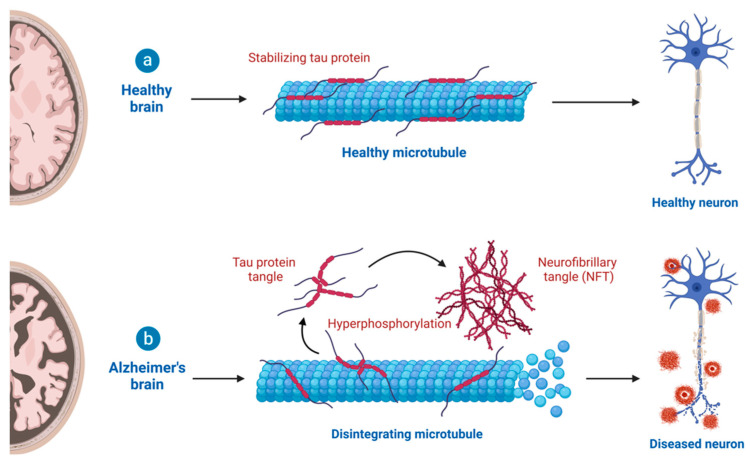
Schematic illustration of tauopathy in AD. Tau protein is usually unfolded and highly soluble. It is responsible for assembly of tubulin and stabilization of microtubules, promoting normal neuronal function. In AD, its randomly coiled structure turns into β-sheet conformation, favoring tau protein misfolding and aggregation. Further, its most common and notable post-translationally modified form is the hyperphosphorylated tau (p-tau). Created with BioRender.com.

**Figure 5 cells-12-01369-f005:**
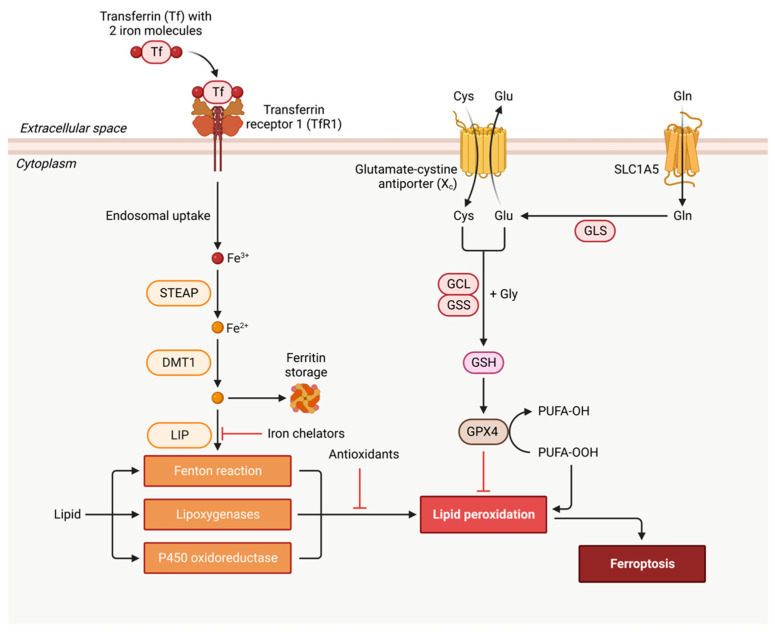
Schematic illustration of ferroptosis signaling pathway. Ferroptosis starts with Fe^3+^-bound transferrin (Tf) endocytosis via transferrin receptor 1 (TfR1). After endocytosis, Fe^3+^ is released from the Tf–TfR1 complex and is reduced to Fe^2+^ by endosomal metalloreductase, STEAP. Fe^2+^ can be transported and stored in ferritin by divalent metal transporter 1 (DMT1 or SLC11A2) or remain in the cytoplasm as a labile iron (Fe^2+^) pool (LIP). Fe^2+^-/Fe^3+^-catalyzed Fenton reaction generates the hydroxyl radical (OH•) that reacts with membrane lipids (the free polyunsaturated fatty acids or PUFA-OH) and produces lipid peroxidation product, PUFA-OOH. Lipid peroxidation can also occur via enzymes such as lipoxygenases (LOXs) such as 15-LOX-1 isoform and P450 oxidoreductase. Ferroptosis can be interdicted by iron chelators, antioxidants, and glutathione peroxidase 4 (GPX4) that converts PUFA-OOH back to PUFA-OH, using glutathione (GSH) as a substrate. GSH synthesis occurs via the entry of cystine into the cell by system X_c_^−^ [36]. Created with BioRender.com.

**Figure 6 cells-12-01369-f006:**
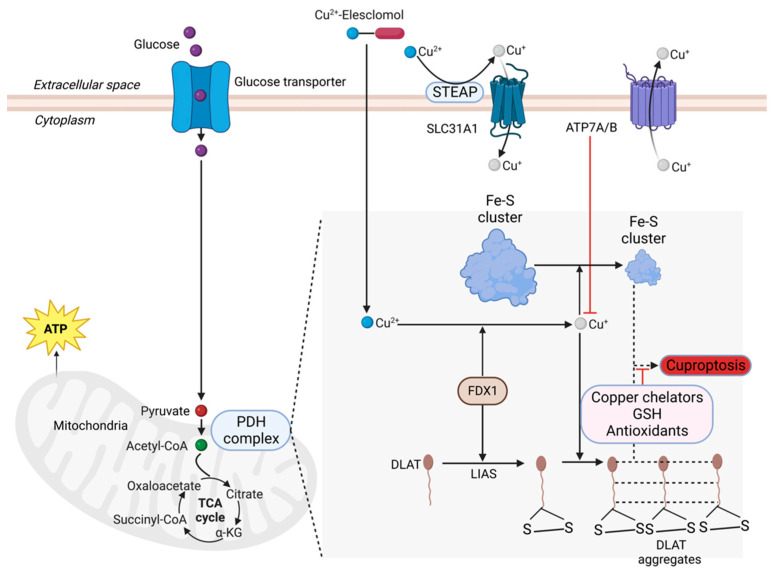
Schematic depiction of cuproptosis mechanism. Excessive extracellular Cu^2+^ is reduced by metalloreductase STEAP to Cu^+^, which is transported into intracellular compartments by SLC31A1 or copper transporter 1 (CTR1). Alternatively, copper ionophores such as elesclomol chelates extracellular Cu^2+^ and enters cells. The key mediators involved in cuproptoosis are: ferrodoxin 1 (FDX1), lipoic acid (LA) synthesis such as lipoyl synthase (LIAS), and protein lipoylation targets such as dihydrolipoamide S-acetyltransferase (DLAT). The DLAT is within pyruvate dehydrogenase (PDH) complex that is linked to mitochondrial tricarboxylic cycle (TCA). On one hand, FDX1 reduces Cu^2+^ to Cu^+^, facilitating the DLAT lipoylation and aggregation. On the other hand, FDX1 causes the destabilization and loss of Fe–S cluster proteins. This leads to proteotoxic stress and cuproptosis [60,61]. Cuproptosis can be inhibited by copper chelators, GSH, and antioxidants. Upregulating copper exporters such as APP7A/B may also be a viable anti-cuproptosis strategy. Created with BioRender.com.

**Table 1 cells-12-01369-t001:** Recent and selected studies on anti-ferroptosis agents as potential AD therapeutics.

Agent/Experimental Model	Mechanism of Action (MOA)	Reference
Plasmalogens (PLs)/AlCl_3_-exposed AD zebrafish	suppressing neuronal ferroptosis and accelerating synaptic transmission at the transcriptional level	[47]
Tetrahydroxy stilbene glycoside (TSG)/APP/PS1 AD transgenic mice	activating GSH/GPX4/ROS and Keap1/Nrf2/ARE signaling pathways, and consequential suppressing of related ferroptosis	[48]
Quinone compounds/HT22 and MC65 cells	inhibiting lipid peroxidation	[49]
Eriodictyol/APP/PS1 AD transgenic mice	interdicting ferroptosis via vitamin-D-receptor-mediated Nrf2 activation	[50]
Forsythoside A (FA)/male APP/PS1 AD transgenic mice, Aβ42-exposed N2a cells, erastin-stimulated HT22 cells, and LPS-induced BV2 cells	inhibiting ferroptosis-mediated neuroinflammation via Nrf2/GPX4 axis activation	[51]
Senegenin (Sen)/PC12 cells	mitigating Aβ25-35-induced oxidative damage and lipid metabolic associated with ferroptosis	[52]
Deferoxamine/rats	ameliorating aluminum-maltolate-(Al(mal)_3_)-induced neuronal ferroptosis and attenuating oxidative stress	[53]

## Data Availability

Not applicable.

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
