# Peer review of "A Concise Review on Oxidative Stress-Mediated Ferroptosis and Cuproptosis in Alzheimer’s Disease"

_cells, 2023, doi:10.3390/cells12101369_

Round 1

Reviewer 1 Report

Excellent content and well organised.

-If author would have suggested animal model designing and/or potential pharmacological targets, that might benefit reader more.

-A figure of ferroptosis and AD relation is suggested

Author Response

“Excellent content and well organised.”

Response: Thanks for the reviewer’s encouragement.

“-If author would have suggested animal model designing and/or potential pharmacological targets, that might benefit reader more.”

Response: Thanks for your suggestions, and I’ve laid out the potential anti-ferroptosis strategies for Alzheimer’s disease, as shown in Figure 5.

“-A figure of ferroptosis and AD relation is suggested.”

Response: Thanks for your suggestions, and I’ve added Figure 5 for this purpose.

Reviewer 2 Report

The here given review provides a good basis, but some explanations are missing, the review should be extended. Brief explanation what ferroptosis and cuproptosis do (e.g. https://doi.org/10.3390/ijms21228765). Maybe two figures would be helpful to better understand the (suggested) mechanism in general and especially in AD.

The role of cupper should be more described. What does it do with Aβ (e.g. https://doi.org/10.1038/s41392-022-01229-y)?

What about the metal ion hypothesis of Alzheimer’s disease?

What about ferroptosis and cuproptosis and tau?

Some more references might be helpful, e.g. line 44, 54 or after 59, line 109

Check for minor spelling and comma errors (e.g. line 55 American – Americans).

Figure 1 -please update, especially quality (really bad).

Use consistent abbreviations (e.g. secretase, SC (line 104/5)

Author Response

“The here given review provides a good basis, but some explanations are missing, the review should be extended. Brief explanation what ferroptosis and cuproptosis do (e.g. https://doi.org/10.3390/ijms21228765). Maybe two figures would be helpful to better understand the (suggested) mechanism in general and especially in AD.”

Response: Thanks for your suggestions, and I’ve added Figure 5 (for ferroptosis) and Figure 6 (cuproptosis) in the text. In addition, I’ve cited the above reference.

“The role of cupper should be more described. What does it do with Aβ (e.g. https://doi.org/10.1038/s41392-022-01229-y)?”

Response: Thanks for your suggestions, and I’ve cited the above reference and described the redox interactions between Aβ and Cu in the text.

“What about the metal ion hypothesis of Alzheimer’s disease?”

Response: Thanks for your suggestions. I limit my review scope to AD-related ferroptosis and cuproptosis since this is only concise.

“What about ferroptosis and cuproptosis and tau?”

Response: Thanks for your suggestions, and I’ve added tau pathology and relevant figures in the text. The roles of tau in ferroptosis and cuproptosis can only be speculated as no direct link has been established.

“Some more references might be helpful, e.g. line 44, 54 or after 59, line 109”

Response: Thanks for your suggestions, and relevant references have been added in the places.

“Check for minor spelling and comma errors (e.g. line 55 American – Americans).”

Response: Thanks for your suggestions, and the manuscript has been checked for potential errors in spelling and grammar.

“Figure 1 -please update, especially quality (really bad).”

Response: Thanks for your suggestions, and the old Figure 1 is now updated as Figure 3.

“Use consistent abbreviations (e.g. secretase, SC (line 104/5)”

Response: Thanks for your suggestions, and I’ve made the requested changes.